# Under-5 Mortality and Its Associated Factors in Northern Nigeria: Evidence from 22,455 Singleton Live Births (2013–2018)

**DOI:** 10.3390/ijerph18189899

**Published:** 2021-09-20

**Authors:** Osita K. Ezeh, Felix A. Ogbo, Anastasia O. Odumegwu, Gladys H. Oforkansi, Uchechukwu D. Abada, Piwuna C. Goson, Tanko Ishaya, Kingsley E. Agho

**Affiliations:** 1School of Health Sciences, Western Sydney University, Locked Bag 1797, Penrith, NSW 2750, Australia; K.agho@westernsydney.edu.au; 2Barmera Medical Clinic (Lake Bonney Private Medical Clinic), Barmera, SA 5345, Australia; f.ogbo@westernsydney.edu.au; 3Translational Health Research Institute (THRI), Campbelltown Campus, Western Sydney University, Penrith, NSW 2571, Australia; 4State Ministry of Health, Jerome Udoji Secretariat Complex, Awka 420218, Nigeria; orbieo77@gmail.com; 5Department of Economics, Nnamdi Azikiwe University, Awka 420218, Nigeria; okwudilijudith@yahoo.com; 6Department of Banking and Finance, Okija Campus, Madonna University, Okija 431121, Nigeria; ucheabada@yahoo.com; 7Department of Psychiatry, College of Health Sciences, University of Jos, Jos 930003, Nigeria; piwunag@unijos.edu.ng; 8Department of Computer Science, University of Jos, Jos 930003, Nigeria; ishayat@unijos.edu.ng

**Keywords:** under-5 mortality, Nigeria Northern Geopolitical Zones, children younger than 5 years, mortality rate, child mortality

## Abstract

The northern geopolitical zones (NGZs) continue to report the highest under-5 mortality rates (U5MRs) among Nigeria’s six geopolitical zones. This study was designed to identify factors related to under-5 mortality (U5M) in the NGZs. The NGZ populations extracted from the 2018 Nigeria Demographic and Health Survey were explored to assess the factors associated with U5M using logistic regression, generalised linear latent, and mixed models. Between 2013 and 2018, the northwest geopolitical zone reported the highest U5MR (179 deaths per 1000 live births; 95% confidence interval [CI]: 163–194). The adjusted model showed that geopolitical zone, poor household, paternal occupation, perceived children’s body size at birth, caesarean delivery, and mothers and fathers’ education were highly associated with increased odds of U5M. Other significant factors that influenced U5M included children of fourth or higher birth order with shorter interval ≤ 2 years (adjusted odds ratio [aOR] = 1.68; CI: 1.42–1.90) and mothers who did not use contraceptives (aOR = 1.41, CI: 1.13–1.70). Interventions are needed and should primarily spotlight children residing in low-socioeconomic households. Educating mothers on the benefits of contraceptive use, child spacing, timely and safe caesarean delivery and adequate care for small-sized babies may also reduce U5M in Nigeria, particularly in the NGZs.

## 1. Background

Under-5 mortality (U5M), which is defined as the death of children aged between 0 and 59 months [1], remains a challenging public health issue in Nigeria. For instance, of the 5.2 million global deaths of children younger than 5 years old in 2019, Nigeria contributed nearly 0.9 million of these deaths, which earned Nigeria the highest contributor to global U5M [1]. Notably, an overwhelming number of these deaths are preventable or treatable illnesses with optimal healthcare and affordable, accessible and cost-effective interventions [2]. In 2017, for example, diarrhoea- and lower respiratory tract infection-related deaths accounted for approximately 13% and 19%, respectively, of U5M in Nigeria [3].

In the past 10 years, the under-5 mortality rate (U5MR) has decreased by approximately 16% from 157 deaths per 1000 live births in 2008 to 132 deaths per 1000 live births in 2018 [4]. This modest improvement may be attributed to increased proportion of educated women and improved access to maternal and child healthcare services, such as delivery assisted by skilled personnel, prenatal, and postnatal care [4]. However, the pace of decline remains very slow, and if this trajectory continues, achieving the sustainable development goal (SDG–3), of reducing child mortality to 25 deaths per 1000 live births or less by 2030 [5] may not be feasible. Despite the national U5MR improvement, there are subnational differences across the six geopolitical zones (North Central, (NC); North East, (NE); North West (NW), South East (SE), South West (SW), and South-South (SS)) in Nigeria, with persisting disparities in the northern geopolitical zones (NGZs) [6]. A recently reported subnational child mortality estimate indicated sharp U5MR differences across the six geopolitical zones, ranging from a maximum U5MR of 187 deaths per 1000 live births (NW) to a minimum U5MR of 62 deaths (SW) [4]. Socio-economic and cultural disparities [7], inadequate access to health care, and differences in public health interventions coverage [8] have been reported to contribute to the uneven U5MR across NGZs. 

Few studies of U5M conducted in the post-millennium development goal (MDG) era at the subnational [9] and national levels [10,11] in Nigeria have shown that individual maternal and child factors (e.g., maternal age, child gender, birth order), socio-economic factors (e.g., education or wealth), and community factors (e.g., rural residence or geopolitical zone location) were related to U5M. One of the major limitations of these studies is a lack of detailed analyses of the NGZ subgroups due to the multifaceted integrated structural and contextual factors that differentially affect U5M across communities in Nigeria. Salau et al. previously suggested that the use of community-level data [12] or regional specific data may yield better estimates for effective interventional policies, which may lead to a substantial reduction of the U5MR. Additionally, using estimates that are generalised among geopolitical zones may be misleading, or can provide inadequate data for interventional designs for other geopolitical zones given differences in cultural, structural and socio-economic improvement within and across Nigeria’s communities [13]. 

For additional improvement in U5M in Nigeria, locally-specific and standardised household studies that focus on identifying risk associated with the death of children younger than 5years old in disaggregated geopolitical populations in Nigeria are crucial. Therefore, pairing NGZs with similar characteristics (i.e., socio-economic, ethnic, cultural and religious beliefs) could unbridle reporting differences and unhampered effective interventions coverage. Notably, no disaggregated population-based studies have assessed the independent risk factors related to U5M in the three combined NGZs (NC, NE and NW), or the population attributable risk proportion (PAR%) for the adjusted associated factors. The PAR% estimates can assist policymakers to appropriately evaluate prevention and intervention costs to ensure optimum economic and health benefits are achieved in the wider NGZs population.

Accordingly, this study examined potential characteristics that influence U5M in the NGZs using population-based data from the 2018 Nigeria Demographic and Health Survey (NDHS). Recorded estimates in this study will guide health policymakers to develop workable and effective tailored evidence-based interventions to substantially reduce U5M in NGZs that will subsequently lead to a remarkable decline in the U5M nationwide.

## 2. Materials and Methods

This study utilised the birth recode file dataset (NGBR7AFL.DTA) from the 2018 NDHS survey. Information relating to a weighted total of 87,877 singleton and multiple live births in the NGZs was reported by women aged 15–49 years old, consisting of 26,293 live births from NE; 39,928 live births from NW; and 21,656 live births from NC. Reported live births ≥5 years and multiple births were excluded from the weighted total, resulting in a total of 22,455 (NC = 4422, NE = 5989, and NW = 12,045) singleton live-born infants of a mother within a 5-year period preceding the mother’s 2018 NDHS interview date, which was employed for the study analyses. Multiple births were excluded from the analyses as a previous study in Nigeria [14] has shown that multiple births have higher mortality odds than singletons. We also excluded the parity variable because it was a combination of live births and stillbirths (≥20 weeks gestation), while this current study focused mainly on singleton deaths.

Data from all three geopolitical zones (NC, NE and NW) were pooled for the analysis of characteristics related to U5M in the NGZs because few singleton deaths were recorded during the 2018 NDHS for each geopolitical zone and more than 80% of the population share a similar culture and religion. Singleton live births were limited to a 5-year period to reduce the recall bias from women who had given birth at different intervals prior to the interview date. The sampling and questionnaire methodology utilised in gathering information concerning births and deaths has been described elsewhere [4].

### 2.1. Outcome Variable

The study’s primary outcome variable was U5M, which is defined as death between birth and 59 months of age. Death occurring within the specified age period was considered dichotomous, such that each death case was coded as “1” and each survived case was coded as “0”.

### 2.2. Potential Mortality Associated Factors

Past studies of U5M [9,10,11,15,16], especially in sub-Saharan African countries, had a major role in the assessment of possible confounding variables. These potential mortality- associated factors were adapted to the information available in the 2018 NDHS and grouped into four categories (community, socioeconomic, individual (maternal and child), and health service factors). Place of residence and geopolitical zone were classified as community-level factors, while household wealth index, maternal education, maternal literacy level, paternal occupation, and paternal education were grouped as socioeconomic-level factors. The economic status of the households interviewed during the survey was measured using a principal component analysis procedure [17]. This procedure was used to determine the weights on household assets (i.e., television, radio, refrigerator, car, bicycle, motorcycle, source of drinking water, type of toilet facility, electricity and types of building materials used in the place of dwelling) to estimate the household wealth index factor score. The household wealth index factor score was categorised into five quintiles (poorest, poorer, middle, richer, and richest) in the 2018 NDHS. However, in the analysis, we recategorised the household wealth index into three classes. The bottom 40% of households were arbitrarily referred to as poor households, the next 40% as middle households, and the top 20% as rich households.

Individual-level factors consisted of maternal characteristics (age, body mass index, birth order, birth interval and wanted pregnancy at the time of pregnancy) and child characteristics (perceived baby size by mothers and gender) were included in the study. The birth weights of actual newborns were not incorporated because approximately 50% of newborns were not weighed at the time of birth [4]. As a result, perceived baby size by mothers (small or very small, and average or large) was utilised instead of actual baby weight at birth.

Health-service-related factors, such as antenatal and postnatal care services, were not considered as this study was based on all singleton births in the 5-year time point preceding the survey interview date. Information regarding antenatal and postnatal care services in the 2018 NDHS was only detailed for the most recent birth (or last birth) in the 5-year period prior to the survey. However, place of birth, mode of delivery, contraceptive use, and type of delivery assistance, categorised as a health professional and non-health professional (i.e., traditional birth attendant, relatives or friends), were included in the study. Two other important variables, religion and ethnicity, were not also considered in the study because the NGZs population is predominantly Muslim (>90%), with the Hausa/Fulani being the major (>80%) ethnic group, and smaller percentages may produce an estimate with a wider confidence interval. Detailed classification of all the variables used for the study analysis are presented in Table 1.

### 2.3. Statistical Analysis

Initially, the ‘syncmrates’ command in STATA, as detailed by Rutstein and Rojas [18], was employed to estimate the U5MR (and the corresponding 95% confidence interval, CI) for all the independent study factors. This was followed by logistic regression generalised linear latent and mixed models in which the logit link and binomial family that adjusts for clustering and sampling weights were utilized for univariable and multivariable analyses to estimate the crude odds ratios (OR) and adjusted odds ratios (aOR), which measure the risk-level associated with the study outcome. 

A manually stepwise backwards elimination approach was applied for the multivariable analysis to identify independent variables that were significantly associated with the study outcome. The following approaches were utilised to verify our manually backward elimination process to minimise statistical bias: (1) in the baseline multivariable model, only potential associated factors with a *p*-value < 0.20 obtained in the univariable analysis were included in the backward elimination procedure; (2) the backward elimination was re-assessed by including all potential independent risk factors; and (3) collinearity was also tested and reported in the final model. Adjusted risk factors associated with the study outcome at a 5% significance level were retained in the model. 

To validate the estimates obtained from the manually backward approach, we also used a hierarchical technique for the multivariable analysis. This means each of the four level factors (community, socioeconomic, individual related, and health related services) was independently examined. In first-stage modelling, community level factors were entered as a base model and those factors that were statistically significant were retained (model 1). In second stage modelling, the socioeconomic variables were added to the significant factors in model 1 and those variables that were significantly significant were retained in model 2. In third stage and fourth stage modelling, a similar approach was used for individual and health-related variables (Appendix A). All analyses were performed using ‘SVY’ commands in STATA/MP V.13.0 (Stata Corp, College Station, TX, USA). 

Adjusted PAR% was obtained to quantify the U5M attributable to each of the identified significant associated factors in NGZs, Nigeria. The PAR% and 95% CI were calculated using previous similar approaches [19,20] to measure the magnitude of the total risk of U5M in the NGZs’ population that was attributable to each significantly related factor that was retained in the study’s final multivariable model. It was obtained using the following formula, PAR% = pr × ((aOR−1)/aOR)) × 100; where pr is the proportion of the population exposed to the significantly associated factors, and aOR was the adjusted OR for U5M.

## 3. Results

In the three combined geopolitical zones (NC, NE and NW), a weighted total of 2348 deaths in children younger than 5 years old occurred (NC = 324, NE = 556, and NW = 1468) during the study period (2013–2018). The observed U5MR for singleton live-born children between 2013 and 2018 in the NGZs was (150 deaths per 1000 live births; 95% CI: 140–160). The U5MR for NW was greater than the U5MR for NC and NE (Figure 1). The U5MRs for children younger than 5 years old whose mothers cannot read was higher than those for mothers who could read parts of sentences or whole sentences (169 vs. 88 deaths per 1000 live births). Children younger than 5 years old whose households were classified as poor had U5MR that was nearly three-fold the U5MRs of rich households (176 vs. 62 deaths per 1000 live births). 

Children younger than 5 years old whose fathers were engaged in agricultural work had a greater U5MR than those whose fathers were either not working or had paid employment. It was also noted that the U5MRs were greater for people with rural residences, mothers or fathers who had no formal education, mothers who did not use any form of contraception, and mothers younger than 20 years of age (Table 2). 

As presented in Table 3, children younger than 5 years old of fathers who had paid employment in the non-agricultural sector (aOR = 1.58, 95% CI: 1.16–2.15) or fathers who had agricultural work (aOR = 1.44, 95% CI: 1.06–1.96) had greater odds of U5M compared with children whose fathers were not working. There were significantly greater odds of U5M for children born to mothers who had no formal education (aOR = 1.32; 95% CI: 1.05–1.67) and mothers who had primary education (aOR = 1.28, 95% CI: 1.02–1.60) compared with children whose mothers had secondary or higher education. Co-linearity investigation revealed that when educational level of the mother was substituted by maternal literacy in the final model, we noted that children whose mothers cannot read a sentence (aOR = 1.20, 95% CI: 1.01–1.44) were more likely to die than children of mothers who could read parts of sentences or whole sentences.

Compared with the NC, the odds of U5M in the NW increased significantly by 36%. Children born to mothers from poor households (aOR = 1.64; 95% CI: 1.14–2.36) were more likely to die, as were children whose fathers had no formal education (aOR = 1.30; 95% CI: 1.12–1.52). A further collinearity check showed that when the household wealth index was replaced with the place of residence in the final model, increased odds of U5M were observed for children born to mothers residing in rural areas. However, it was statistically insignificant (aOR = 1.10, 95% CI: 0.95–1.29). Other characteristics that posed significant higher odds of U5M included children whose birth size was perceived as small or smaller, children born to mothers who did not use any form of contraception, and children delivered via caesarean section. In addition, there was an increased likelihood of U5M among second or third-born children with two years interval or less, and 4th or higher birth order born children with an interval greater than two years (Table 3). 

As shown in Table 4, the PAR% estimates indicated that, in all the three geopolitical zones included in this study, the proportion of U5M attributed to a lack of use of any form of contraception was 27% (PAR%: 0.27; 95% CI: 0.11–0.41) under the assumption that this relationship was casual. Similarly, the proportion of U5M attributed to poor households was 27% (PAR%: 0.27; 95% CI: 0.08–0.42). 

## 4. Discussion

Findings from the extracted NGZs populations from the 2018 NDHS nationwide representative survey showed that the U5MRs in the three NGZs are still very high and need further interventions to close the gap in mortality rates among the geopolitical zones in the country. During the study period (2013–2018), we observed sharp variations of U5MR across the NGZs, ranging from a low rate of 102 deaths per 1000 live births (NC) to the highest rate of 179 (NW). The observed U5MR for the NGZs was (150 deaths per 1000 live births), which is well above the most recent reported U5MR (132 deaths per 1000 live births) [4]. The study further identified that geopolitical zone (NW), household wealth index (poor household), maternal education (no formal or primary education), paternal education (no formal or primary education) and paternal occupation (engaged in non-agricultural or agricultural work) were significantly associated with higher odds of U5M. Contraceptive use (non-use), baby size at birth (small or very small), caesarean delivery, and birth order with birth interval (second or third-order children with shorter birth intervals (≤2) years or fourth or higher birth order with a short birth interval (≤2) years) were also identified to be highly associated with increased likelihood of U5M in the NGZs.

As noted in this study, the risk of U5M was significantly higher in the NW geopolitical zone compared with the NC and NE geopolitical zones. This outcome reaffirms earlier studies on the relationship between geopolitical zone variations and U5M in Nigeria [11,21]. The geopolitical differentials observed may be linked to economic resource inequality and disparities in social development, and inadequate access to health facilities and health personnel. However, it is surprising that the NW continues to report the highest U5MR [4], although government interventional programs were successfully established and implemented in the NW states (Jigawa, Katsina, and Zamfara) in the past two decades. These interventions included the Partnership for Reviving Routine Immunization in Northern Nigeria (PRRINN), which was established in 2006; in 2008, PRRINN was expanded to include maternal, newborn, and child health (PRRINN-MNCH) [22] whose core aims include reduction of U5M. However, evidence from the NDHS report on childhood mortality showed that between 2013 and 2018, the NW U5MR rose by 1.1% from 185 deaths per 1000 live births in 2013 to 187 in 2018 [4], indicating that intervention policies implemented since 2006 were unproductive. Therefore, the current outcome indicates an urgent need for further research on NW geopolitical populations to unravel other potential factors that may be contributing to U5M. Mortality odds for children younger than 5 years old residing in poor households was significantly greater than the mortality odds for those in rich households. This outcome contradicts a similar study conducted in Nepal in 2018, which indicated a statistically insignificant relationship between the household economic status and the U5M [23]. However, our current finding is similar to the findings of earlier studies [24,25]. Our finding could be attributed to a recent report that found that approximately 40% (or 83million people) of the total population live below US$1 (or NGN420) per day [26], and approximately 87% of all the poor in Nigeria are concentrated in the NGZs [27]. This poverty constraint impacts mothers in many ways, including gaining access to maternal healthcare services at well-equipped health facilities and residing in an environment with good water and sanitation infrastructure. The constraint also affects mothers’ health-seeking behaviour, resulting in a high likelihood of U5M. This study reaffirms the need for the three tiers of government (local, state, and federal) to retool their poverty intervention programs, particularly those in conflict-infected areas, to reduce U5M.

Fathers’ occupations, particularly those in non-agricultural paid employment and those engaged in agricultural work, were indicative of significantly increased odds of U5M compared with non-working fathers. This finding contradicts similar studies conducted in Ethiopia [28,29], which showed that fathers who were engaged in any type of paid employment had a protective effect on U5M. There are a few other studies that have also reported an insignificant relationship between U5M and fathers’ occupational status [30,31]. These differing reports may be linked to different classifications of occupation, potential risk factors adjusted for their studies, and different populations. Nevertheless, Fenta et al. also indicated in their study that working fathers had a greater likelihood of child mortality than non-working fathers [32]. The significantly elevated U5M related to working fathers of NGZs in Nigeria may be attributed to inadequate time to care for their children, which often results in untimely healthcare-seeking behaviours for their children.

It was also observed that children whose fathers and/or mothers had no formal education had 1.30 times greater odds of U5M and 1.32 times greater odds of U5M, respectively, compared with those who had secondary or higher education. These findings were similar to studies carried out in Bangladesh [33], Chad [31] and Tanzania [25]. A study conducted in Kenya indicated an insignificant relationship between mothers’ education level and U5M [34]. A plausible explanation of our finding could be that uneducated mothers are less likely to understand the benefits of antenatal and postnatal care, and good childcare practices (i.e., hygienic behaviours, immunization, preventative care, suitable and timely feeding). This finding is supported by a study performed in Ethiopia in 2010, which revealed that a substantial number of women do not promptly seek care because of a lack of understanding of the benefits that positively impact child survival [35]. Uneducated mothers are more likely to engage in socio-cultural practices that may negatively impact child survival; for example, cultural prelacteal feeding practices that deprive newborns of colostrum—a fluid rich in nutrients and immunoglobulins [36]. Nationally, the reported U5MR for uneducated mothers was 170 deaths per 1000 live births between 2013 and 2018 [4], which is similar to the current study findings for the NGZs. This finding reaffirms that women’s empowerment through education remain a powerful key strategy that would remarkably reduce U5M in Nigeria, particularly in the NGZs where culture and religious belief is still rife against women (e.g., force/child marriages, inheritance discrimination, and female genital mutilation). Children of educated women are more likely to receive better childcare at home and appropriate healthcare when due [37]. Therefore, supporting women’s education and skills acquisition training will not only reduce U5M but it will empower women economically and socially to fend for themselves and their families, increase their self-confidence and have control over health resources [38].

Findings from this study showed that children younger than 5 years old whose mothers did not use contraception had significantly greater odds of dying in the U5M period than those who had access to contraceptives. Similar results were detailed in previous studies carried out in Kenya [39], Bangladesh [40], and Nepal [23]. In addition, a recent study in Nigeria opined that mothers who used contraceptives had significantly lower U5M [41]. Fear of side-effects due to contraceptive use may be attributable to the non-use of contraceptives among NGZs mothers, resulting in the increased likelihood of U5M noted in our study. For instance, previous studies in Nigeria have suggested that the alleged fear of side-effects, which include womb damage, menstrual irregularities, delay in return to fertility, partner objection, and difficulties in breastfeeding contribute to the lack of interest in contraception [42,43]. Another plausible reason for the observed increase in U5M among mothers who did not use contraception may be cultural beliefs or religious practice. Previous studies have suggested that mothers who practice Islam were less likely to use contraceptives [44,45]. The effect of non-use of contraceptives among mothers remains an amendable associated factor for U5M. Whilst misinformation has largely driven the fear of side-effects of contraceptive use, highlighting the need for effective public health interventions (i.e., educating mothers and their partners/husband on the benefits of contraceptive use, intensifying media campaigns on adverse effects of high-risk births and providing free family planning essentials at primary health facilities) to scale up contraceptive use is crucial in reducing high-risk births and child mortality.

Children of birth order (two through four or higher) born with a shorter birth interval of two years or less had a significantly greater likelihood of U5M than those with a longer birth interval (>two years). This finding is in line with previous cross-sectional studies conducted in India [46], Kenya [47], Bangladesh [40], and Tanzania [25]. However, our findings contradict a recent similar study conducted in Benin, which suggested that fourth or higher-order children with an interval greater than two years were significantly associated with U5M [10]. An increased odds of U5M observed may be attributable to inadequate care and attention given to higher ordered children and insufficient economic resources, which often leads to competition among siblings, especially in poorer households, and adversely impacts maternal health and well-being [48]. Another possible contributing factor may be obstetric complications which are higher among mothers with shorter birth intervals than those with longer birth intervals [49].

Furthermore, the odds of U5M were significantly greater for children delivered by caesarean section than those delivered via vaginal birth, which was consistent with previous findings [24,25]. However, our findings contradict earlier studies conducted in Swaziland [50] and Egypt [51] which reported an insignificant relationship between caesarean delivery and child mortality. Negative perceptions (i.e., fear, dislike and misconception) of caesarean section delivery [52] among women in the NGZs may have contributed to the significantly higher odds of U5M noted in the study. This finding may explain why pregnant women present for emergency caesarean section after experiencing life threatening complications of labour elsewhere [53], and Ezechi et al. in their study on pregnant Nigerian women’s view of caesarean section reported that over three-quarters of child deaths occurred in pregnant women who had emergency caesarean deliveries [54]. However, caution needs to be exercised in interpreting this finding as elective and emergency caesarean delivery was not discerned in the information recorded in the 2018 NDHS.

Children who were perceived as small- or very-small-sized after birth by their mothers had a higher likelihood of U5M than those perceived as average- or larger-sized. This result contradicts a study performed by Yaya et al. [10]; however, few other similar studies showed a significant relationship with U5M [25,55,56,57,58]. Even if our outcome on perceived children’s size was significant, the rationale mothers applied in estimating their children’s size after birth remains unclear. Therefore, caution should be exercised in making inferences with this result, as preterm and small for gestational age infants were not differentiated in the 2018 NDHS. A study in Bangladesh has suggested that approximately three-quarters of deaths associated with low birth weight in children were attributed to a preterm condition [59].

It has been suggested that the PAR% estimates are very crucial when the exposure factor is casually associated with the outcome and the exposure factor is amenable to strategic benefits [60]. However, in our cross-sectional study we assumed that the relationship was casual. The PAR% findings indicated that among women who did not use any form of contraception, the interventions which is an amenable factor may have wider population reduction on total U5M among NGZs population and a similar reduction on the U5M among children from poor households which is also a modifiable factor. Non-use of contraception and poor households may likely offer substantial marginal benefit than the other associated factors in the study. The NGZs population PAR% estimates can assist policy makers in developing effective prevention and intervention costs to ensure adequate allocation of resources according to geopolitical and/or states health and economic needs.

### Strengths and Limitations

The strengths of this current study are presented as follows: (1) the study restricted births within a 5-year period to reduce recall bias (related to dates of birth and death) and potential changes in household characteristics (i.e., economic status); (2) a population-based study, with over 80% in same ethnic group (with the same culture, religion, economic and social lifestyles), increases the validity of the estimates; (3) the 2018 NDHS data satisfied the conditions for calculating PAR; for example, events of interest and exposure information were randomly selected and recorded with lessened bias; and (4) this study specifically indicated NGZs evidence on key characteristics related to U5M, which will inform tailored intervention programs to reduce child deaths in the three geopolitical zones (NC, NE, and NW).

Limitations of this study included: (1) it is possible that U5M may have been under-reported as only surviving mothers participated in the survey; (2) information regarding maternal medical condition, such as infection, diabetes and hypertension prior to childbirth or during childbirth was not considered in the study as it was lacking in the 2018 NDHS, which may have impacted our estimates. Additionally, maternal depression after birth earlier found to be associated with U5M in Taiwan [61] was not included due to unavailability of data; (3) medical status of the children (i.e., birth asphyxia, jaundice and sepsis) and causes of death were not available for investigation; (4) Actual birth weight of newborns after birth, which may be a crucial factor related to U5M, was not investigated in this study because at least 50% of the newborns were not weighed after birth. The mother’s perceived newborn size at birth used in place of actual birth weight may have affected U5M odds estimates reported because rationale or criteria mothers used was unclear. The mother’s newborn size assessment could be limited by their prior knowledge and experience of newborns; (5) PAR% was calculated based on the assumption that there exists a causal relationship between the identified risk characteristics in the study and U5M in the NGZs; (6) it is possible that investigated factors (e.g., mode of delivery) may have biased our findings because emergency and elective caesarean section were combined during the 2017/18 NDHS data collection; and (7) The World Health Organization body mass index classification cut-off used for adolescent women aged 15–18 years may have led to over- or underestimation of estimates.

## 5. Conclusions

Findings from the investigated characteristics related to deaths in children younger than five years old in the NGZs indicated that NW geopolitical zone, poor household, fathers engaged in paid work in the non-agricultural or agricultural sector, perceived children body size at birth by their mothers, caesarean delivery, mothers or fathers who had no formal education, a higher birth order preceding interval ≤two years and mothers who did not use contraception reported a statistically significantly higher likelihood of U5M in the NGZs. Interventions are needed and should focus on children residing in low socioeconomic households and geopolitical zones with inadequate access to primary health care services. Educating mothers on the benefits of contraceptive use, child spacing, well-timed safe caesarean delivery, and adequate care for small-sized babies may substantially help in scaling down U5M in Nigeria, particularly in the NGZs.

## Figures and Tables

**Figure 1 ijerph-18-09899-f001:**
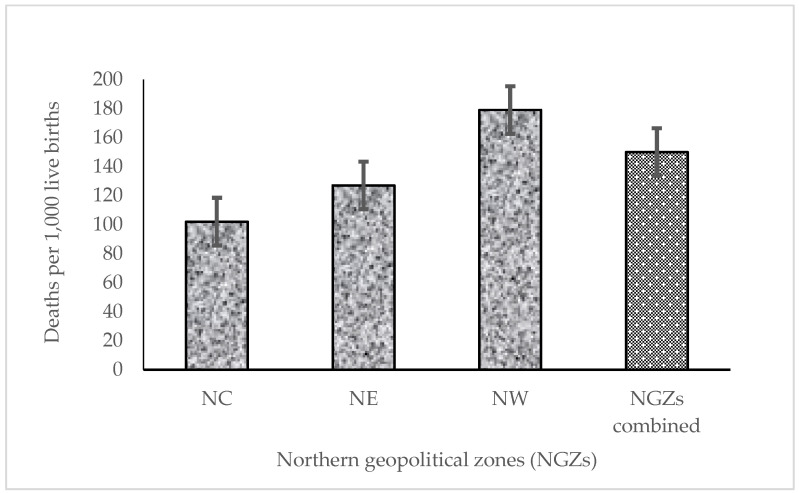
Under-5 mortality rate per 1000 live births (singleton) by Northern geopolitical zones, Nigeria, 2013–2018. NC, Northcentral; NE, Northeast; NW, Northwest.

**Table 1 ijerph-18-09899-t001:** Definition and classification of independent variables used for the study analysis.

Independent Variables	Classification
Community level factor	
Residence	1 = urban; 2 = rural
Geopolitical zone	1 = North Central; 2 = North East; 3 = North West
Socioeconomic factor	
Household wealth index	1 = Poor; 2 = Middle; 3 = Rich
Maternal educational level	1 = No education; 2 = Primary; 3 = Secondary or higher
Maternal literacy level	1 = Able to read parts of & whole sentence; 2 = Cannot read at all
Father’s educational level	1 = No education; 2 = Primary; 3 = Secondary or higher
Father’s occupation	1 = Not working; 2= Non-agricultural work; 3 = Agricultural work
Number of women in the Household	1 = One woman; 2 = At least 2 women
Individual related factor	
Mother’s age at child birth (years)	1 = < 20; 2 = 20–29; 3 = 30–39; 4 = 40–49
Mother’s body mass index (kg/m^2^) (MBMI)	1 = Normal (18.5 ≤ MBMI ≤ 24.9); 2 = Underweight (MBMI < 18.5); 3 = Overweight (25 ≤ MBMI ≤ 29.9); 4 = Obese (MBMI ≥ 30)
Wanted pregnancy at the time of pregnant	1 = Wanted then; 2 = Wanted later; 3 = Wanted no more
Child sex	1 = Female; 2 = Male
Mother’s perceived baby size	1 = Average or Larger; 2 = Small or very small
Birth order and birth interval	1 = Second or third child, interval > 2 years; 2 = First child; 3 = Second or third child, interval ≤ 2 years; 4 = Fourth or higher child, interval > 2 years; 5 = Fourth or higher child, interval ≤ 2 years
Health related factor	
Contraceptive use	1 = Yes; 2 = No
Place of birth	1 = Home; 2 = Health facility
Mode of delivery	1 = non-caesarean; 2 = caesarean section
Delivery assistance	1 = Health professional; 2 = non-Health professional

kg, weight measured in kilograms; m^2^, height measured in square meters.

**Table 2 ijerph-18-09899-t002:** Under-5 mortality rate, and distribution of characteristics reported in the 2018 NDHS, in three combined Northern geopolitical zones, Nigeria.

Variable	*n*	*n*%	* U5MR (95% CI)
Community-level factor			
Residence type			
Urban	5818	490 (21)	109 (94–125)
Rural	16,637	1858 (79)	164(151–177)
Geopolitical zone (North)			
North Central	4422	324 (14)	102 (85–118)
North East	5988	556 (24)	127 (112–143)
North West	12,045	1468 (62)	179 (163–194)
Socioeconomic level factor			
Household wealth index ⱡ			
Rich	1555	82 (3)	62 (43–81)
Middle	7508	652 (28)	120 (105–134)
Poor	13,392	1614 (69)	176 (162–190)
Mother’s education			
Secondary or higher	4825	293 (12)	77 (66–87)
Primary	3036	302 (13)	136 (120–152)
No education	14,594	1754 (75)	175 (162–187)
Mother’s literacy level			
Able to read	5535	379 (16)	88 (78–98)
Cannot read	16,920	1969 (84)	169 (157–181)
Father’s education			
No education	11,933	1382 (62)	183 (169–196)
Primary	2632	279 (13)	148 (129–168)
Secondary or higher	7627	551 (25)	99 (87–111)
Father’s occupation			
Not working	1753	156 (7)	120 (90–149)
Non-agricultural work	10,980	1100 (47)	138 (124–152)
Agricultural work	9722	1092 (46)	169 (149–188)
Number of women in the household			
One woman	12,226	1198 (51)	143 (123–162)
At least 2 women	10,229	1150 (49)	158 (147–170)
Individual related factor			
Mother’s age			
<20	1225	148 (6)	186 (146–226)
20–29	11,382	1165 (50)	149 (130–167)
30–39	7686	767 (33)	142 (130–155)
40–49	2162	268 (11)	161 (140–182)
Mother’s body mass index (kg·m^2^) (MBMI) ¥			
Normal (18.5 ≤ MBMI ≤ 24.9)	14,745	1506 (65)	146 (134–159)
Underweight (MBMI < 18.5)	4828	521 (23)	158 (142–175)
Overweight (25 ≤ MBMI ≤ 29.9)	2166	241 (11)	155 (124–185)
Obese (MBMI ≥ 30)	34	37 (1)	131 (47–214)
Wanted pregnancy at the time of pregnant			
Wanted then	21,073	2228 (95)	151 (141–162)
Wanted later	1085	97 (4)	119 (87–151)
Wanted no more	297	23 (1)	119 (62–177)
Mother’s perceived baby size ¥			
Average or larger	19,056	1893 (81)	144 (133–155)
Small or very small	3247	436 (19)	185 (156–213)
Sex			
Female	11,065	1106 (47)	148 (124–172)
Male	11,390	1242 (53)	152 (137–166)
Birth order/birth interval			
First	3924	469 (20)	165 (131–199)
2nd or 3rd rank, interval ≤ 2 yrs	2072	289 (12)	179 (148–209)
2nd or 3rd rank, interval > 2 yrs	4759	360 (15)	108 (95–120)
4th or higher rank, interval ≤ 2 yrs	8402	714 (30)	134 (122–146)
4th or higher rank, interval > 2 yrs	3298	516 (22)	202 (180–224)
Health-related factor			
Contraceptive use			
Yes	2130	133 (6)	85 (65–104)
No	20325	2215 (94)	157 (146–168)
Place of birth			
Home	16,972	1902 (81)	162 (150–174)
Health facility	5483	446 (19)	109 (92–125)
Mode of delivery ¥			
Non-caesarean	22,186	2314 (99)	150 (140–160)
Caesarean ‴	243	34 (1)	179 (111–240)
Delivery assistance ¥			
Health professional	5871	457 (20)	106 (91–121)
Non-health professional ^	16,042	1831 (80)	165 (152–178)

*n* (%) Weighted number, and proportion of children <5 years old who died between 0 and 59 months; * All U5MR estimates with 95% confidence interval (CI) were rounded to a whole number; ⱡ Wealth was assessed based on household assets (radio, television, fridge, bicycle, motorcycle, car, telephone, electricity, and type of floor material used in rooms); NDHS (Nigeria Demographic and Health Survey); ^ Traditional birth attendant, relative or friend; ‴ Caesarean section is a combination of both elective and emergency caesarean; yrs Years; ¥, variables with missing values (MBMI = 337, mode of delivery = 26, mother perceived baby size at birth = 152, father’s education = 149, delivery assistance = 449); yrs, years; kg, weight measured in kilograms; m^2^, height measured in square meters.

**Table 3 ijerph-18-09899-t003:** Crude and adjusted odd ratios 95% confidence interval for factors related to under-5 mortality NGZs, Nigeria.

Variable	Crude Model ^†,§^	*p*-Value	Adjusted Model ^†,§^	*p*-Value
Community level factor				
Residence type				
Urban	Ref			
Rural	1.35 (1.14–1.60)	<0.001	―	―
Geopolitical zone				
North Central	Ref		Ref	
North East	1.30 (1.08–1.55)	0.005	1.06 (0.88–1.27)	0.563
North West	1.71 (1.45–2.01)	<0.001	1.36 (1.14–1.61)	0.001
Socioeconomic level factor				
Household wealth index +				
Rich	Ref		Ref	
Middle	1.73 (1.22–2.45)	0.002	1.42 (0.99–2.03)	0.056
Poor	2.37 (1.71–3.30)	<0.001	1.64 (1.14–2.36)	0.008
Mother’s education				
Secondary or higher	Ref		Ref	
Primary	1.52 (1.24–1.86)	<0.001	1.28 (1.02–1.60)	0.034
No education	1.92 (1.61–2.30)	<0.001	1.32 (1.05–1.67)	0.018
Mother’s literacy level				
Able to read	Ref			
Cannot read	1.71 (1.47–1.99)	<0.001	―	―
Father’s education				
Secondary or higher	Ref		Ref	
Primary	1.48 (1.23–1.78)	<0.001	1.20 (0.99–1.45)	0.051
No education	1.73 (1.51–1.99)	<0.001	1.30 (1.12–1.52)	0.001
Father’s occupation				
Not working	Ref		Ref	
Non-agricultural work	1.37 (0.99–1.87)	0.052	1.58 (1.16–2.15)	0.004
Agricultural work	1.51 (1.11–2.06)	0.009	1.44 (1.06–1.96)	0.020
Number of women in household				
One woman	Ref			
At least 2 women	1.17 (1.04–1.31)	0.009	―	―
Individual related factor				
Mother’s age				
<20	1.60 (1.24–2.05)	<0.001	―	―
20–29	1.08 (0.97–1.21)	0.152	―	―
30–39	Ref			
40–49	1.21 (1.02–1.42)	0.027	―	―
Mother’s body mass index (kg/m^2^) (MBMI)				
Normal (18.5 ≤ MBMI ≥ 24.9)	Ref			
Underweight (MBMI< 18.5)	1.06 (0.94–1.19)	0.394	―	―
Overweight (25 ≤ MBMI ≥ 29.9)	1.07 (0.88–1.29)	0,457	―	―
Obese (MBMI≥ 30)	0.85 (0.53–1.37)	0.573	―	―
Wanted pregnancy at the time of pregnant				
Wanted then	Ref			
Wanted later	0.83 (0.64–1.09)	0.178	―	―
Wanted no more	0.71 (0.44–1.14)	0.151	―	―
Mother’s perceived baby size				
Average or larger	Ref		Ref	
Small or very small	1.44 (1.22–1.69)	<0.001	1.34 (1.16–1.55)	<0.001
Sex				
Female	Ref			
Male	1.07 (0.95–1.22)	0.277	―	―
Birth order/birth interval				
First	1.63 (1.39–1.91)	<0.001	1.62 (1.39–1.90)	<0.001
2nd or 3rd rank, interval ≤ 2 yrs	1.82 (1.45–2.30)	<0.001	1.79 (1.43–2.23)	<0.001
2nd or 3rd rank, interval > 2 yrs	Ref		Ref	
4th or higher rank, interval ≤ 2 yrs	1.17 (1.01–1.36)	0.033	1.07 (0.92–1.24)	0.374
4th or higher rank, interval > 2 yrs	1.92 (1.63–2.27)	<0.001	1.68 (1.42–1.98)	<0.001
Health related factor				
Contraceptive use				
Yes	Ref		Ref	
No	1.87 (1.53–2.29)	<0.001	1.41 (1.13–1.75)	0.002
Place of birth				
Home	1.36 (1.17–1.58)	<0.001	―	―
Health facility	Ref			
Mode of delivery				
Non-caesarean	Ref		Ref	
Caesarean ^‡^	1.58 (0.93–2.69)	0.093	2.68 (1.55–4.63)	<0.001
Delivery assistance				
Health professional	Ref			
Non-health professional ^	1.40 (1.21–1.61)	<0.001	―	―

^†^ a logistic regression generalized linear latent and mixed models was used to estimate the odd ratio with 95% confidence interval; ^§^ 1113 missing values were not included in the model; ^‡^ Caesarean section is a combination of both elective and emergency caesarean; ^ Traditional birth attendant, relative or friend; Ref Reference category; yrs years; + Wealth was assessed based on household assets (radio, television, fridge, bicycle, motorcycle, car, telephone, electricity, and type of floor material used in rooms); NGZs, Northern geopolitical zones; kg, weight measured in kilograms; m^2^, height measured in square meters.

**Table 4 ijerph-18-09899-t004:** Estimated PAR% for each of the factors significantly related to under-5 mortality in the NGZs, Nigeria.

Variable	*n*% *	aOR ‡	PAR% (95% CI)
Geopolitical zone			
North Central	0.138	Ref	
North East	0.237	1.06	―
North West	0.625	1.36	17 (7–25)
Household wealth index			
Rich	0.035	Ref	
Middle	0.278	1.42	―
Poor	0.687	1.64	27 (8–42)
Mother’s education			
Secondary or higher	0.125	Ref	
Primary	0.128	1.28	3 (0–6)
No education	0.747	1.32	18 (3–31)
Father’s education			
Secondary or higher	0.249	Ref	
Primary	0.126	1.2	―
No education	0.625	1.3	14 (6–23)
Father’s occupation			
Not working	0.066	Ref	
Non-agricultural work	0.468	1.58	17 (6–27)
Agricultural work	0.465	1.44	14 (2–25)
Mother’s perceived baby size			
Average or larger	0.813	Ref	
Small or very small	0.187	1.34	5 (2–8)
Birth order/birth interval		Ref	
First	0.2	1.62	8 (5–11)
2nd or 3rd rank, interval ≤ 2 yrs	0.123	1.79	5 (3–8)
2nd or 3rd rank, interval > 2 yrs	0.153	Ref	
4th or higher rank, interval ≤ 2 yrs	0.304	1.07	―
4th or higher rank, interval > 2 yrs	0.22	1.68	9 (6–12)
Mode of delivery			
Non-caesarean	0.986	Ref	
Caesarean	0.014	2.68	1 (0–2)
Contraceptive use			
Yes	0.057	Ref	
No	0.943	1.41	27 (11–41)

* Weighted proportion of under-five deaths. ‡ The adjusted model included the place of residence; geopolitical zone; household wealth index; mother’s (marital status, education, age, body mass index, desire for pregnancy); father’s education; father’s occupation; the number of women in the household; child sex; place of birth; delivery assistance; mode of delivery; child’s body size at birth; birth order and birth interval and contraceptive use. aOR, adjusted odds ratio; ― PAR was not calculated due to insignificant relationship to U5M; CI, Confidence interval; Ref, reference category; yrs, Years; NGZs, Northern geopolitical zones.

## Data Availability

This study was based on a public domain dataset that is freely available online. https://dhsprogram.com/data/dataset/Nigeria_Standard-DHS_2018.cfm?flag=0 (accessed on 4 December 2020).

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
