# Peer review of "Under-5 Mortality and Its Associated Factors in Northern Nigeria: Evidence from 22,455 Singleton Live Births (2013–2018)"

_ijerph, 2021, doi:10.3390/ijerph18189899_

Round 1

Reviewer 1 Report

Dear author,

The paper entitled " Under-5 mortality and its associated factors in Northern Nigeria: evidence from 22,455 singleton live births (2013-2018) is truly fascinating. Under 5 mortality remains one of the biggest challenges in the LMICs and they need further interventions to close the gap in mortality rates in different countries. I have some comments for the paper that needs to be addressed before accepting for publication.

  1. Line 53: What are the reasons for the declined pace of U5MR? Please add.
  2. Please add information about the covariates that were adjusted during the analysis.
  3. In the result section, please clarify the category of a wanted pregnancy. What does wanted no more mean? Is it unwanted pregnancy or the mother desired for baby but once she conceived, she did not want it anymore.
  4. Was parity of mothers considered in this study? Several studies have associated high parity births with higher child mortality than low parity births. If it is not considered, then it should be added to the limitations and properly discussed.
  5. The low-birth-weight infant remains at a much higher risk of mortality than the infant with normal weight at birth. The authors mentioned that around 50% of newborn's birth weight was not available. This should be added to the limitations.

Author Response

Dear Reviewer,

We have attached response to Reviewer comments point by point in a tabular form attached herewith.

Reviewer 2 Report

Thank you for the opportunity to review this peace.

The paper's goal was to investigate associated factors with under 5-years mortality rates in Northern regions of Nigeria, a country with high rates of the problem globally.

The authors did a good job when using traditional strategies to seek variables associated with the outcome. However, I suggest the usage of different approaches that I will explain below.

Overall, the work is of good quality as it was based on a large dataset of information of national and regional representativeness.

My main point is to not refer to your results as “risk.” You don’t have longitudinal data, even though it is possible to establish temporality for some of the information. Most of your data were collected concomitantly with the information about child death. I recommend using “odds were higher,” for example.

Abstract:

Line 30: Please, add the unit always when referring to U5MR (per 1000 live births, percentage, etc.). This is valid throughout the paper.

Material and Methods:

Add to the covariates if they were used as continuous or categorical in the models. If categorically, also add the categories explored. I only realize how the variables were investigated when checking the Tables.

Line 124: Some variables may present collinearity, such as maternal education x maternal literacy level and paternal occupation x paternal education. You described that tests for collinearity were employed, but I couldn’t find any appropriate consideration in the results. So I ask you to clarify how collinearity as testes and when it occurred, how did you deal with, please.

Line 130: The maternal BMI as it is provided in DHS datasets was used in the analysis. I presumed that it was used according to WHO’s cut-offs for all women, adults and adolescents. If yes, I think it is a wrong way to use this variable, even seeing the same approach in other papers. For adolescent mothers, the BMI-for-age z-scores must be used to classify their nutritional status. Additionally, give more details about how you used the BMI in the analysis.

Concerning the backward process, I strongly suggest using a hierarchical approach to select variables into the adjusted models based on a previously established model of determinants of the outcome.

Results:

Line 172: Same comment as in the Abstract. Add the unit to all U5MR.

Line 176: Please, give more details on how you classified the household as poor and rich. Was it based on splitting the wealth index into tertiles? Moreover, the DHS wealth index classifies the household wealth, not the individuals, as poor or rich. Therefore, the mortality rate was higher among children living in the poorest households compared to the richest ones.

Line 198: You described that proportions varied due to missing values. However, I did not find any mention in the text about the inspection of missing values and how they can affect the findings of your work.

Line 203: Same comment as previously. Most of your data do not allow the investigation of risk (longitudinal) but associated factors instead (cross-sectional).

Table 2: Do not leave blank spaces in the adjusted column, please. Add a hyphen or results, if available.

Lines 223-225: I do not understand this result. Why have you replaced the wealth index?

Discussion:

Line 255: Is the average U5MR ideal to report? Wouldn’t the median be more adequate?

Line 271-273: Although your work aimed to explore factors associated with U5MR, I think it is crucial to investigate other rates, such as neonatal and infant mortality. By stratifying the analysis, the reasons for not reducing U5MR can be better addressed.

Line 321: Please, consider the role of women’s empowerment in the association with child mortality.

Line 336: Why haven’t you considered religion affiliation a potentially associated factor with U5MR in the analysis?

Line 387: One of my suggestions is to consider including ethnicity as a potential variable in the analysis, based on this recent paper by Victora et al. (https://www.thelancet.com/action/showPdf?pii=S2214-109X%2820%2930025-5). However, you well stated that more than 80% of the people living in the Northern regions of Nigeria belong to the same ethnic group. In this sense, I ask you to add some sentences in the Introduction or Methods to justify why this variable was not considered in the analysis.

Author Response

Dear Reviewer 2,

Please find attached response to your comments.

Thanks

Reviewer 3 Report

Thanks for the opportunity to review the manuscript. It has an interesting research question, although my main concern is related to the novelty of the study. There is another one with similar approach used here already published, which used the MICS survey in Nigeria (ref #39). Also, the results are quite similar to this paper.

The paper seems to have an adequate data analysis and the conclusions are supported by the results.

Specific points that need to be addressed:

1) introduction should also mention the importance of PAR for policy makers

2) Methods should be more detailed, specially when variables were described. One example is maternal education. Authors should clearly state that they used categories of maternal education or years of formal schooling. This appears as Tables footnotes, but should be in the methods.

3) Data presentation in tables should present p-values. Although I agree with authors that confidence intervals are sufficient, the authors stated in the methods that selection of variables in the model are based on p-values. So, this information is important to be in the tables. The text also need improvement, specially when report that something is associated to the mortality. When say this, direction and magnitude of the association should be stated.

4) Discussion should be more detailed. Two main points: a) authors presented some results but not discussed them (as PAR); b) authors mentioned they are surprised by one region presenting the highest mortality rate even when this region is the target for some policies, but mentioned relatively new policies and these policies could not impact on mortality yet. The time between the policy implementation and impact on mortality should be discussed.

Minor points are presented in the PDF file attached to this review.

Author Response

Dear Reviewer 3,

Please find attached response to your comments.

thanks

Round 2

Reviewer 3 Report

Authors provided reasonable answers to my comments, although we disagree in some points, especially in the comment that the old #39 reference (now #41) use a similar analytical approach besides use the region as variables and the survey used was the MICS. Results are quite similar to those observed here. Also, the authors should include p-values for variables, not for the categories of each variable, as it is written in the methods section as a criteria to keep variables in the model (lines 177-187).

Author Response

Dear Reviewer 3,

Please find attached response to your comments.

Thanks
